# The Role of Intestinal Microbiota and Dietary Fibre in the Regulation of Blood Pressure Through the Interaction with Sodium: A Narrative Review

**DOI:** 10.3390/microorganisms13061269

**Published:** 2025-05-30

**Authors:** Agnieszka Rudzka, Dorota Zielińska, Katarzyna Neffe-Skocińska, Barbara Sionek, Aleksandra Szydłowska, Karolina Górnik-Horn, Danuta Kołożyn-Krajewska

**Affiliations:** 1Department of Dietetics and Food Studies, Faculty of Science and Technology, Jan Dlugosz University in Czestochowa, Al. Armii Krajowej 13/15, 42-200 Czestochowa, Poland; k.gornik-horn@ujd.edu.pl (K.G.-H.); d.kolozyn-krajewska@ujd.edu.pl (D.K.-K.); 2Institute of Human Nutrition Sciences, Department of Gastronomic Technology and Food Hygiene, Faculty of Human Nutrition, Warsaw University of Life Sciences, Nowoursynowska 159 C, 02-776 Warsaw, Poland; dorota_zielinska@sggw.edu.pl (D.Z.); katarzyna_neffe-skocinska@sggw.edu.pl (K.N.-S.); barbara_sionek@sggw.edu.pl (B.S.); aleksandra_szydlowska@sggw.edu.pl (A.S.)

**Keywords:** hypertension, sodium, fibre, gut microbiota, sodium bioavailability

## Abstract

Sodium consumption is a well-established risk factor for the development of hypertension. Nevertheless, current recommendations for reducing dietary sodium intake are challenging to implement. Consequently, alternative approaches that would reduce the harmful health effects of excessive sodium consumption on blood pressure are highly desirable. The scientific evidence suggests that dietary fibre intake and human intestinal microbiota may affect blood pressure regulation, potentially through interactions with sodium. This narrative review aims to explore the complex interactions between microbiota, fibre, and the fate of sodium in the human body, as well as the implications of these interactions in the prevention and treatment of hypertension. The relevant literature, published up to March 2025, was searched across databases including Google, Google Scholar, PubMed, and Web of Science. A total of 160 most relevant references were included. Gathered evidence suggests that while dietary fibre may reduce sodium uptake into the bloodstream by binding sodium ions and bile salts, microbiota may also contribute to lowering sodium bioavailability thanks to multiple metabolites with anti-inflammatory and intestinal sodium transporter-modulating properties. Despite these promising findings, further work is needed to allow the translation of these insights into effective therapeutic approaches, particularly for salt-sensitive, hypertensive individuals.

## 1. Introduction

Hypertension is a condition and a serious risk factor for the development of many diseases. Very well-known consequences of this condition are cardiovascular diseases and kidney failure, but scientific evidence also links it to dementia, cancer, osteoporosis and oral diseases [1]. In addition, it is estimated that hypertension shortens life expectancy by about 5 years [2]. Despite these consequences, hypertension is a prevalent pathological condition. Worldwide, 1.28 billion people aged 30–79 suffer from hypertension [3], which stands for approximately 16% of the total population. The economic costs of healthcare for the hypertensive population vary among countries between 1.3 and 8.5% of gross domestic product [4]. Hence, hypertension is an important problem affecting the healthcare system worldwide, which emphasises the importance of actions limiting its occurrence and intensity.

Known risk factors for the development of hypertension, apart from age, genetics, and low level of physical activity, include diet, especially high consumption of sodium and alcohol [3]. A newly acknowledged cause as well as the consequence of hypertension is the dysbiosis of intestinal microbiota [5]. Studies show that the faecal microbiota of hypertensive and normotensive people differs in the composition. In hypertension, the relative abundance of bacteria of the genus *Klebsiella*, *Prevotella*, and *Enterobacter*, increases at the expense of short-chain fatty acid (SCFA) producers such as *Fecalibacterium*, *Ruminococcus*, *Roseburia*, *Bifidobacterium*, *Bacteroides*, and *Akkermansia* [6]. In addition, it has been shown that the α-diversity of the microbiota is negatively correlated with the value of systolic blood pressure and hypertension [7]. This effect was observed, even though in the cited study, 153 out of 183 hypertensive patients were taking antihypertensive drugs, which in the vast majority of cases (128 people) allowed an adequate control of blood pressure. Hence, the antihypertensive drugs do not eliminate the microbiota dysbiosis associated with hypertension. Experts indicate, however, that the microbiota may impact the action of such drugs, and hence the selection of an appropriate treatment could benefit from considering its interactions with the microbial community inhabiting digestive tract of a particular individual [8].

Given the common occurrence of dysbiosis in hypertensive people, interventions that bring out the health-promoting properties of the microbiota could help to tackle the problem of hypertension. One such intervention is the change in dietary habits that would allow for nourishing the microbial community that dwells in the intestines. Every micro and macro nutrient reaches the colon in some part, but the largest amount of residues constitutes dietary fibre. A high fibre intake is recommended in the DASH (Dietary Approaches to Stop Hypertension) [9] diet due to the positive effect of this nutrient on blood pressure regulation. The DASH diet is based on fruits and vegetables, wholegrains, low-fat dairy, fish and poultry, as well as low amounts of red meat. It limits the intake of processed foods, especially ultra-processed items, such as sugar-based beverages, that are consumed in large volumes in Western countries [10].

While the influence of microbiota and dietary fibre on blood pressure regulation is well-documented, the precise mechanisms remain underexplored. This review aims to succinctly summarise the mechanisms through which the fibre and microbiota may impact blood pressure, focusing on their interactions with sodium—a key dietary factor in the pathogenesis of hypertension.

## 2. Methodology

The literature search included peer-reviewed articles and governmental or organisational websites presenting current dietary recommendations for sodium intake, published online in English up to March 2025. While we have focused on recent references (<5 years), some older sources of fundamental meaning to the field or unique research value were also included.

The search was conducted using the following databases: Google, Google Scholar, PubMed, and Web of Science, with a combination of the following keywords: “sodium”, “microbiota”, “microbiome”, “dietary fibre”, “diet”, “hypertension”, “sodium sensitivity”, “recommendations for sodium intake”, and “bioavailability of sodium”. In total, over a thousand titles were screened for suitability, and finally, 160 references were used as a basis for this article.

## 3. Blood Pressure in Relation to Sodium Content in the Diet: Current Recommendations

### 3.1. Dietary Sodium Consumption in Relation to Health

The requirement for sodium depends on age, physical activity, and ambient temperature. Its content in the diet should complement the losses with sweat, urine, and faeces, and in the case of children and adolescents, also ensure the proper growth of the body. More physical activity and increased ambient temperature result in greater sodium losses with sweat, and then the demand for this element may be greater [11].

Before the discovery of salt’s food preservation properties by the ancient civilization, the only source of sodium for humans was food. It is estimated that daily sodium consumption was below 0.2 g/day, equivalent to 0.5 g salt (2.5 g of salt comprises 1.0 g sodium) [12]. At present, only primitive tribes, such as Brazilian Yanomami Indians, consume very little sodium. Therefore, the wide range of human adaptations to sodium intake should be considered in dietary recommendations of salt consumption. The current dietary intake of salt is estimated worldwide as 10.78 g/day (from 6.7 g/day in the WHO Africa Region to 15.6 g/day in the WHO Western Pacific Region) [13]. Hence, nowadays, the consumption of salt is more than twenty times higher than in ancient civilizations. Excessive sodium consumption, along with impaired renal sodium excretion, leads to an increased blood pressure level. As a result, the peripheral vascular resistance is elevated. Anatomic remodelling of small resistance arteries is accompanied by vascular endothelial inflammation and structural changes in large elastic arteries (increase in arterial stiffness) [14,15]. In addition to vascular dysfunction, the pathophysiology of sodium-sensitive blood pressure includes many factors, such as genetic, neuro-endocrine (renin–angiotensin–aldosterone system, natriuretic peptides, etc.), and autonomic nervous system imbalance (increased sympathetic activity) [16].

High-sodium diets may increase blood pressure in healthy individuals [17,18]. In turn, reducing dietary sodium intake appears to have a hypotensive effect only in those with relatively high blood pressure (above the third quartile of the population) [19]. People vary in how their blood pressure responds to sodium intake. If the blood pressure increases in parallel with dietary sodium content, then a person is considered sodium sensitive. Sodium sensitivity affects 25% of normotensive and 30–60% of hypertensive individuals [20,21]. Furthermore, populations more at risk of dietary sodium sensitivity include women, the elderly, people with insulin resistance, the obese, those with chronic kidney disease, low weight at birth, and African American or Asian descent [22,23]. Interestingly, the amount of sodium excreted does not differ between subjects with and without sensitivity to dietary sodium, but a consistent increase in blood pressure occurs only in the former [20]. Sensitivity to dietary sodium, even in non-hypertensive individuals, is itself a risk factor for cardiovascular morbidity and mortality [24].

The beneficial health effect of lowering blood pressure by reducing salt in the diet was for the first time reported in 1948 by Kempner, who assessed the effect of the rice and fruit diet on patients with hypertension [25]. The majority of subsequent studies confirmed population health benefits of reduced salt intake. According to He et al., the reduction of 2.5 g of daily salt consumption is associated with a 20% reduced risk of cardiovascular events [12]. It should be emphasised that lower salt intake is complementary to other non-pharmacological interventions, such as a low-fat diet and weight loss [26]. The results of the study of Bibbins-Domingo et al. showed that a daily salt reduction by 3 g in US population can result in a reduction in annual number of new cases of coronary heart disease by 60,000, stroke by 32,000, and myocardial infarction by 54,000. The yearly reduction in deaths from any cause was projected to decrease by 44,000. Authors underlined that all population segments benefit from reduced salt consumption, and this intervention improves public health along with the reductions in tobacco use, obesity, and cholesterol levels [27]. However, recently it was suggested that very low salt consumption can be harmful. Some confounding data and analyses are showing the relation between sodium intake and cardiovascular events appears as a J-curve [28]. In the study of Mente et al. of 133,118 individuals (63,559 with hypertension and 69,559 without hypertension) the reduction in daily sodium below 3 g resulted in an increased risk of cardiovascular events and death in both hypertensive and normotensive people [29]. In this regard, further studies and meta-analysis are needed.

Despite this concern, the European Society of Cardiology (ESC)/European Society of Hypertension (ESH) guidelines (2024) recommend “to restrict total dietary sodium intake to approximately 2 g/day or less (equivalent to approximately 5 g or about a teaspoon of salt per day). This includes added salt and salt already contained in food”. This is class IA recommendation (class I—evidence and/or general agreement that a given treatment or procedure is beneficial, useful, and effective; level of evidence A—data derived from multiple randomised clinical trials or meta-analyses) [28].

International Society of Hypertension Global Hypertension Practice Guidelines (2020) also recommends salt reduction stating: “There is strong evidence for a relationship between high salt intake and increased blood pressure. Reduce salt added when preparing foods, and at the table. Avoid or limit consumption of high-salt foods such as soy sauce, fast foods and processed food including breads and cereals high in salt” [30].

Currently, the main sources of sodium in the diet in European populations, besides table salt, are: bread, processed meat, and cheese, or additionally in the diet of US residents-confectionery (cookies, brownies, cakes) [31,32]. Small amounts of this element are found in unprocessed foods and in drinking water [32,33].

### 3.2. Nutritional Societies Recommendations of Sodium Intake

The recommendations for sodium intake, issued by different nutritional societies consider various population groups. Besides adults, sodium intake values are guided for infants, children, and adolescents, taking into account such factors as age, growth rate, body weight, height, energy needs, or consumption of milk [34,35,36]. In addition, many documents propose sodium intake levels for elderly, pregnant, and lactating women, which are the same as for the general adult population [34,35,36,37]. The sodium intake values for adults as recommended by several nutritional societies are summarised in Table 1.

In summary, sodium intake seen as adequate or sufficient is similar regardless of the issuing organisation (from 1.5 to 2 g/day). This guided intake is less than a half of a global mean intake for adults (estimated at 4.31 g/day [41]). Therefore, regulatory bodies aim to reduce the sodium intake in the population, which can contribute to improving public health. An example of such regulations is Regulation (EU) No 1169/2011 [42] of the European Parliament and of the Council, which requires consumers to be informed about the salt content of food products. This allows consumers to consciously choose products with a lower sodium content, which promotes healthier eating habits. In addition, some countries have introduced limits on the maximum salt content of certain foods and educational programmes [43,44] to raise awareness of the effects of excessive sodium intake. Such legislative and educational measures are necessary to address the health problems associated with excessive sodium intake effectively. Studies imply that additional countermeasures may also be required. For example, some authors have suggested that addressing the issue of sodium sensitivity through dietary approaches may be more feasible than reducing the intake of salt [45]. They have critically evaluated governmental strategies to reduce the salt concentration in foods, carried out up-to-date, and showed that these had very little effect on the actual salt consumption. Instead, they have recommended DASH diet (where the intake of sodium should not exceed 2.3, or even 1.5 g per day) and the inclusion of potassium chloride and nitrate to prevent vasodysfunction in sodium-sensitive individuals and consequently ameliorate the increase in blood pressure. Worth noting here is that the DASH diet lowers blood pressure on its own as well as in combination with decreased salt intake, giving then even better effects [46]. This positive result cannot be attributed to richness in potassium, magnesium and fibre alone [47] (recommended intake of each of these nutrients in the DASH diet is a minimum of 4.7 g, 0.5 g, and 30 g per day, respectively [9]). Other factors, such as high levels of nitrates and calcium (consumed at a minimum of 1.25 mg per day) that characterise this diet, could also contribute to its hypotensive properties [45]. In addition, the quality of fibre included in the DASH diet might be a very important factor.

## 4. The Effect of Dietary Fibre on Blood Pressure

A 2022 meta-analysis of observational studies and randomised clinical controlled trials on people with cardiovascular disease or hypertension showed with high confidence that increasing fibre intake causes a decrease in systolic and diastolic blood pressure regardless of the use of cardioprotective drugs [48]. This effect is now widely recognised and thus, recommendations for fibre intake to improve blood pressure control were recently suggested at >28 g/day for women and 10 g more for men [49].

Indeed, the increase in foods high in fibre could have a hypotensive effect, but high-fibre dietary regimes differ in effectiveness. Based on recent research, the most effective diets are DASH and lacto-ovo vegetarian diets, while Mediterranean, Nordic, fruit and vegetable, and high-fibre diets do not have as pronounced effects [50]. This brings attention to the adequate choice of dietary fibre optimal for hypertension.

Many studies where precise kinds of fibre were looked at, indicated that not only the quantity but also the quality of dietary fibre is important for its hypotensive effects. A detailed review of the literature prepared by Aleixandre and Miguel (2016) showed that diets and supplements containing, in particular, soluble fibre caused a reduction in blood pressure [51]. The hypotensive effect has been demonstrated in human studies for fibres, preparations and products such as oat fibre, including beta-glucan [52,53,54,55], psyllium [56], grape antioxidant fibre product (75% fibre and 18.7% polyphenols) [57], soluble fibre from Plantago ovata husks [58], cocoa husk based supplement [59], lupin flour enriched bread (high fibre and protein) [60], guar gum [61,62], sesame [63], flaxseed [64], chia seeds [65], chitosan [66], and seasoning made of grape pomace [67]. On the other hand, some fibres did not impact blood pressure values. This was demonstrated for inulin-type fructans supplemented in humans [68] and cellulose (insoluble fibre) in rodents [69]. Figure 1 summarises the key findings on the hypotensive effects of dietary fibre.

It has been proposed that the mechanism by which soluble fibre lowers blood pressure is the reduction in insulin resistance and insulin blood levels, which when in excess, promote the retention of sodium in the body [70]. In addition, the reduction in blood cholesterol, which is also linked to soluble fibre intake, helps to regulate vasodilation [71,72]. However, researchers admit that the mechanisms through which the fibre can exert hypotensive effects remain underexplored [70].

One of the other possible mechanisms through which the fibre may aid the health of hypertensive individuals is its impact on the microbiota. Soluble fibre is a major fibre fraction influencing the microbiota composition in a beneficial way [73]. Some of the fibre fractions constituting a significant proportion of the dietary residues, such as resistant starch, arabinogalactans and xylans, are partly soluble; however, they still stimulate the microbiota in a positive way [74,75]. For example, resistant starch increases the abundance of such bacterial taxa as *Clostridium*, *Butyricoccus*, *Faecalibacterium*, *Prevotellaceae*, *Ruminococcaceae*, *Bifidobcterium*, *Akkermansia*, *Blautia*, *Eubacterium*, *Roseburia*, *Rombutsia,* and *Caprococcus* [75]; xylan is a good source of food for *Bifidobacterium* and *Lactobacillus* [74], arabinogalactan for *Roseburia* [76], and pectins present in fruits and vegetables encourage proliferation of *Bacteroietes*, *Clostridiales*, *Prevotella*, *Porphyromonas*, *Bifidobacterium*, *Olesnella*, *Lactobacillus Lachnospiraceae*, *Facealibacterium,* and *Paracbacteroides* [76,77]. Different fibre fractions impact the quantity and quality of SCFA produced by microbiota. According to in vitro research performed by Bai et al. (2021), resistant starch allows microbiota to produce high concentrations of butyrate, citrus pectin-propionate, and xylan-acetate [74]. Furthermore, the experimentation on male Wistar rats showed that arabinogalactan is a better substrate for butyrate, propionate, acetate, and valerate production when compared to apple pectin, xylan, guar gum, carrageenan, glucomannan, β-glucan, xylan, and xanthan gum [76]. Further studies on possible hypotensive effects of fibre–microbiota interaction are indispensable, since so far, a major part of the scientific evidence points to the hypotensive effects of butyrate [74].

Alleged mechanisms involved in hypotensive effects of dietary fibre are summarised in Figure 2.

## 5. The Influence of Gut Microbiota on Blood Pressure

The literature has repeatedly described a significant relationship between food consumption and the composition of the gut microbiota, and increasing evidence also supports the role of the gut microbiota in the regulation of blood pressure [78]. Studies conducted in both animal models and humans have shown a strong correlation between changes in the composition of the gut microbiota and the development of hypertension. By influencing blood pressure, the gut microbiota may potentially influence the progression of cardiovascular and kidney diseases [79].

Animal models of hypertension, including spontaneously hypertensive rats and induced hypertension rats, have been found to have altered gut microbiota structure compared with their normotensive strains [80]. Gut microbiota dysbiosis has also been observed in studies on hypertensive human patients, mainly conducted in Chinese, American, and Brazilian populations. For example, studies conducted in a population of 41 healthy individuals, 56 hypertensive patients, and 99 patients with primary hypertension in northern China demonstrated significantly reduced microbial diversity and dysbiosis of microbial function in patients with prehypertension and hypertension [81]. In contrast, differences in gut microbiota were observed between White and Black individuals in the American population. Black individuals with hypertension were found to have higher blood pressure, a higher prevalence of treatment-resistant hypertension, stronger proinflammatory properties of the gut microbiota, and more markers of oxidative stress than White individuals with hypertension [82].

The gut microbiota may contribute directly to the pathogenesis of hypertension. There are several reviews focusing on the association between gut microbiota and hypertension [82,83,84]. However, it is important to define which are the potential microorganisms regulating blood pressure at the taxonomic level and which probiotics may play a role in order to treat hypertension. A recent review cited several studies, e.g., on the correlation between hypertension and 18 different bacterial genera, including *Anaerovorax*, *Clostridium* IV, *Oscillibacter,* and *Sporobacter*, and that the presence of *Veillonella* in particular was consistent in patients with hypertension [79]. Furthermore, *Anaerovorax*, *Catabacter,* and *Robinsoneilla* were positively correlated with hypertension [79]. In contrast, in a Finnish cohort study of 6953 individuals, 45 genera of microorganisms were observed to be positively associated with blood pressure, of which 27 belonged to the phylum *Firmicutes* [85]. In another study, the gut microbiota of 60 Chinese patients with hypertension was identified based on an increased number of opportunistic pathogens, such as *Klebsiella*, *Streptococcus,* and *Parabacteroides* [86]. In contrast, the number of SCFA producers, including *Roseburia* and *Bacillus freundii*, was reduced. Additionally, it has been shown that among hypertensive patients, statistically significant lower gut microbiota diversity occurs [7]. On the other hand, microbial richness and evenness remain largely unaffected by the hypertensive status [83].

The mode of action of the intestinal microbiota in hypertension can be both beneficial and adverse. Microbiota metabolites such as SCFA and indole-3-lactic acid (beneficial) trimethylamine N-oxide (unfavourable) have been shown to interact with gene pathways important in regulating blood pressure [87] and also directly affect the renin–angiotensin–aldosterone system through, e.g., the production of enzyme inhibitors or angiotensin converting enzymes [88]. The mechanism of action by which gut microbiota affects blood pressure and contributes to the development of hypertension includes changes in gut barrier function, gut microbiota structure, and gut microbial metabolites.

Imbalance of the gut microbiota can lead to gut barrier dysfunction and increased intestinal permeability. This increases the risk of pathogenic bacteria and their associated lipopolysaccharides (LPS) entering the bloodstream, causing systemic inflammation. These changes can further lead to endothelial cell dysfunction and vascular sclerosis, causing or exacerbating hypertension. In addition, gut barrier dysfunction can impair the growth of beneficial bacteria, leading to gut microbiota imbalance [89]. The proliferation of Gram-negative bacteria leads to increased LPS levels, which then bind to Toll-like receptor 4 (TLR4) to activate the innate immune response. LPS forms a complex with LPS-binding protein, enters the blood from the intestine, interacts with CD14 on monocytes, and activates TLR4, thereby triggering the production of proinflammatory cytokines such as tumour necrosis factor alpha (TNF-α), interleukin-1 (IL-1), and IL-6 [90].

The mechanisms by which gut microbiota imbalance causes hypertension include several aspects. The unbalanced gut microbiota usually manifests as a reduced number of beneficial bacteria and an increased number of harmful bacteria. It has been shown that in the case of hypertensive patients, the number of harmful *Proteobacteria* is significantly increased, which causes the gut microbiota structure imbalance. That kind of dysbiosis is closely related to intestinal inflammation and immune dysfunction. Moreover, the unbalanced gut microbiota leads to the abnormal expression of tight junction proteins, zonula occludin-1 (ZO-1), and occludin in the intestinal mucosa, thereby increasing intestinal permeability and impairing intestinal barrier function. As this condition persists, pathogenic bacteria in the intestine and enterotoxins increase. Harmful substances enter the bloodstream through the mesentery, inducing chronic inflammation and vascular endothelial damage, decreasing vasodilator factors, and increasing vasoconstrictor factors. Increased peripheral resistance ultimately causes increased blood pressure [91,92].

In addition, the gut microbiota can stimulate intestinal chromaffin cells to produce serotonin, dopamine, and norepinephrine, which could also influence blood pressure levels. The microbiota–gut–brain axis facilitates bidirectional communication between the human gut microbiota and itself. It can be characterised as interactions that involve the gut-associated immune system, the enteric nervous system, the vagus nerve, and the gut microbiota that secretes neurotransmitters, tryptophan, and short-chain fatty acids. This pathway is essential for human health, but the specific mechanisms underlying the influence of the axis on hypertension are not fully understood [93].

The gut microbiota can also regulate blood pressure by modulating steroid hormone levels. Certain gut microorganisms can metabolise mineralocorticoids and endogenous glucocorticoids as well as some of their respective derivatives. Many of these steroidal metabolites are reabsorbed back into the circulation and may be biologically active, influencing blood pressure [94].

It was also proved that the gut microbiota produces many metabolites, such as trimethylamine-N-oxide (TMAO), SCFAs, corticosterone, hydrogen sulphide (H2S), choline, bile acids (BAs), indole sulphate, LPS, etc., among which SCFAs, TMAO, and LPS are closely related to the development of hypertension [88].

Studies have shown that SCFA-producing gut microbiota is associated with reduced blood pressure. On the other hand, elevated faecal SCFA level is associated with increased blood pressure. The most important role in blood pressure regulation is played by the G-protein-coupled receptors (mainly GPR41 and GPR43) and olfactory receptor 78 (Olfr78), which are SCFA receptors. SCFAs, by acting on these receptors, exert anti-inflammatory effects, therefore regulating gut microbiota and influencing blood pressure levels [95].

The oxidation of trimethylamine (TMA) by gut microbiota causes the TMAO production. However, the metabolic pathways differ in different parts of the gut and are dependent on the microbiota composition. It has been shown that plasma TMAO can accelerate foam cell formation, increase oxidative stress and proinflammatory responses, and reduce the production of anti-inflammatory cytokines. It was observed that when TMAO levels increase, the intestinal permeability and inflammation in the body also increase [84,96].

Modulation of the gut microbiota through dietary interventions and probiotics has shown promise in regulating blood pressure and reducing systemic inflammation, offering a new approach to treating hypertension. For example, the Mediterranean diet, which is rich in polyphenols and omega-3 fatty acids and low in sodium, promotes the growth of beneficial gut bacteria that support cardiovascular health. In addition, probiotics have been found to enhance gut barrier function, reduce inflammation, and modulate the renin-angiotensin system, all of which contribute to lower blood pressure [79].

## 6. Dietary Fibre and Microbiota as Key Factors Influencing Sodium Homeostasis in the Body

Dietary fibre is a compound able to modulate the bioavailability of both macro and micronutrients [97]. For example, research by Jimoh et al. (2016) showed that the mucilage from psyllium husk was able to trap at least 50% of sodium, and the level of retention during digestion in in vitro conditions simulating the stomach and intestine did not decrease significantly [98]. However, the authors of the mentioned study did not take into account the influence of the microbiota on fibre integrity and sodium release, and the simulation focused only on maintaining the physiological pH of the digestive tract.

Nevertheless, the microbiota may significantly affect the bioavailability of micronutrients entrapped in the fibre matrix. Firstly, because the microbiota feeds on fibre and produces SCFAs, that may improve the solubility of some minerals, and secondly, because the elements can also be absorbed in the large intestine where the microbiota is most abundant. In the case of sodium, the large intestine is the main part of the digestive tract responsible for its absorption, and hence, both fibre and microbiota may be particularly important for sodium’s bioavailability.

In vitro studies have shown that fibre binds elements in its structure, and, depending on its quality, to a different degree [99]. However, in in vivo studies, on humans and animals, a decrease in the bioavailability of the elements with an increased supply of fibre was very often not observed, and some authors reported quite the opposite effect [100]. Therefore, the inclusion of the microbiota in in vitro studies to assess the effect of dietary fibre on the bioavailability of minerals, especially sodium, is very important. An even better alternative is the use of in vivo research where the microbiota is always present.

For example, the sodium-binding properties of psyllium husk have been confirmed by some in vivo studies. The early work of Obata et al. (1998) demonstrated that male, stroke-prone spontaneously hypertensive rats increased the faecal excretion of sodium when fed a control diet with 3 or 10% psyllium husk compared to when given a control diet with or without the addition of 10% cellulose [69]. The authors also found that the inclusion of psyllium husk in the rats’ diet resulted in a lower ventricular weight, possibly due to amelioration of hypertension through the increased excretion of sodium. This property of psyllium husk was further confirmed in humans [101]. When healthy Japanese men supplemented psyllium husk, they excreted more sodium with the faeces. At the same time, the supplementation did not seem to affect faecal levels of potassium, calcium, magnesium, phosphorus, copper, zinc, manganese, and iron.

Other than psyllium husk, authors also looked at the impact of wheat fibre on the faecal excretion of sodium in rats [102]. In the mentioned study animals were fed a wheat bran cereal-based diet (28% of dietary fibre), standard rat chow (5% of dietary fibre), and an elemental diet with no fibre. The faecal sodium content increased with the increase in fibre in a diet (from 16 to nearly 2000 μ equivalents). The dietary sodium intake could partly affect this result (the authors did not disclose whether it was constant across the applied diets); however, it is likely that such a big difference was predominantly due to the binding of sodium by the fibre. In addition to the increased faecal sodium, a high-fibre diet resulted in greater potassium (from 43 to 798 μ equivalents) and faecal water (from 0.18 to 2.76 g) excretion. These results were obtained despite the intestinal hypertrophy in rats fed a high-fibre diet and the apparent increase in mucosal Na^+^/K^+^-ATPase in the ileum, colon, and cecum.

The faecal excretion of sodium may be elevated through the binding of bile salts that contain sodium [103]. Bile salt binding is a known and, from a health perspective, desirable effect of dietary fibre. Faecal excretion of bile salts encourages the liver to produce bile de novo using cholesterol as a substrate. Hence, this process typically triggers a decrease in serum cholesterol levels [104]. However, the increased excretion of bile salts may also impact the fate of sodium in the organism, including its absorption. The reabsorption of bile salts from the small intestine’s lumen is accompanied by the absorption of sodium ions through apical sodium-dependent bile acid transporters [105,106]. One bile salt molecule is transported together with two sodium ions. Considering that the total circulating daily pool of bile acids weighs 12–18 g [107], the sodium used in its reabsorption could amount to approximately 1.3–1.9 g (considering that the weight of two sodium ions is 10.6% of the molecular weight of sodium cholate). This amount would contribute significantly to the circulating sodium levels, taking into account that diet, depending on the salt content, delivers between 1.5 and 4.7 g of this electrolyte daily [9,108]. Therefore, dietary fibre, through the entrapment of bile salts, could limit the uptake of sodium from the intestinal lumen.

Perhaps the most widely recognised fibre fraction with bile-binding capacity is β-glucan [109]. Oat and barley β-glucan have such a well-documented cholesterol-lowering effect that in 2011, the European Food Safety Authority endorsed the use of related health claims on products with the use of these fibres [110]. Beyond β-glucan, the bile-binding ability was demonstrated for a variety of other fibres, such as fibres sourced from sugarcane, amaranth, coconut residue, seaweed, as well as pectin, chitosan, and lignins [103,111,112,113,114,115]. A 2021 meta-analysis found that bile acid excretion in animals was related to the dietary content of overall soluble dietary fibre (as well as the content of protein and fat, but not cellulose) [116].

In summary, the hypotensive effects exhibited by soluble dietary fibre could be partly explained by the ability of these compounds to increase the excretion of bile and sodium with faeces. Nevertheless, the effect of fibre on sodium bioavailability remains an underexplored area of research [56]. In particular, little is known about whether and how dietary fibre may affect the sodium transporters in the intestines. A previously mentioned study on rats showed that dietary fibre increased the number of sodium transporters in the intestines (mucosal Na^+^/K^+^-ATPase) [61]. This could be due to changes in the microbiota caused by this fibre. Research has shown that microbiota-derived SCFA increase the proliferation of epithelial cells [117]. Thus, it might not be surprising that SCFA exposure was also linked to the increase in intestinal sodium transporter (NHE3) expression and the consequent increase in sodium uptake in animal and human cell studies [118]. This mechanism could partly explain the results of a recently published human trial where in response to 4-week butyrate supplementation and an increase in dietary sodium intake, daytime 24 h systolic and diastolic blood pressure of hypertensive patients increased significantly (also when compared to the hypertensive placebo group) [119]. The authors pointed out that the observed increase in blood pressure could result from greater sodium retention in a butyrate-treated group. Nevertheless, the absorption of sodium was not measured in the study.

On the other hand, in another human trial where a prebiotic acetylated and butyrylated high-amylose maize starch was supplemented to hypertensive patients for 3 weeks, a clinically relevant decrease in 24 h systolic blood pressure was noted [120]. These results highlight that fibre intake, rather than SCFA supplementation alone, may be more important for blood pressure regulation.

Interestingly, a group that studied the effect of SCFA exposure on sodium absorption in the omasal epithelium found that SCFA, especially at pH < 7.0, activated the transport of sodium ions but in the longer term suppressed the expression of sodium transporters NHE2 and NHE3 [121]. Further research should explore whether chronic exposure to elevated SCFA levels resulting from fibre intake and microbial activity might downregulate the uptake of sodium in human intestines.

Supplementation with SCFA does not seem to affect the microbiota [119], unlike the increase in dietary fibre intake. Several studies have shown that microbiota may affect sodium transporters in the intestines. Pathological conditions typically accompanied by dysbiosis, e.g., colon cancer, display a lowered expression of NHE3 [122]. Furthermore, the activity of some microbes was directly linked to impaired sodium transport. For example, toxins produced by *Vibrio cholerae,* enterotoxigenic *Escherichia coli,* and *Clostridium difficile* inhibit NHE3 transporters [123]. Interestingly, *C. difficile*-infected humans and NHE3-deficient mice are characterised by reduced Firmicutes and increased Bacteroidetes phyla abundance [124,125]. Conversely to this observation, the decrease in Firmicutes to Bacteroidetes ratio is traditionally thought of as a beneficial change in the microbiota [126]. Furthermore, an increase in this ratio is associated with hypertension [127], and hence it could be considered a therapeutic target.

The inhibition of NHE3 sodium-mediated intestinal transport is a basis for some developing anti-hypertensive drugs. For example, NHE3 inhibitor SAR 218034 was shown to decrease blood pressure in obese spontaneously hypertensive rats through the increase in faecal excretion of sodium [128]. Nevertheless, the medication also had a laxative effect, which is most likely due to the impairment of NHE3 action. Chronic diarrhoea was repeatedly observed in animals with selective deletion of this sodium transporter [129]. Currently, in USA another NHE3 inhibitor, called Tenapanor is approved and available for use [130]. It is, however, not utilised as a drug for hypertension but recommended in constipation and irritable bowel syndrome, as well as hyperphosphatemia associated with chronic kidney disease [130,131]. Exploring the potential of microbiota in the regulation of NHE3-mediated sodium transport might be of interest for hypertension therapy.

Another mechanism through which the intestinal microbiota may impact the homeostasis of sodium in the body is the influence on the tightness of the intestinal barrier. A properly functioning intestinal barrier is crucial for stopping toxins and pathogens from penetrating the human body and regulating the bioaccessibility of nutrients [132]. On the other hand, studies indicate that the dysbiosis accompanying hypertension impairs barrier function and increases intestinal inflammation [133]. Despite this, there is very little literature on the impact of the intestinal microbiota on the bioaccessibility of the key micronutrient for hypertension, specifically sodium. There is also a lack of human studies demonstrating the impact of microbiota intervention on sodium levels in the human body. However, such studies exist for other micronutrients. It has been shown that consuming probiotics or prebiotic substances can positively affect the nutritional status of Fe, Ca, Zn, Cu, and Mg, as well as vitamins D, A, and B12 in various population groups [117,134,135,136,137,138,139,140,141,142,143,144,145]. The microbiota is thus expected to influence, also and perhaps above all, the absorption of sodium in the body since 90% of sodium is absorbed in the large intestine [146] (as opposed to most other nutrients), and this is where the microbiota is most abundant [147].

On the other hand, it is known that the level of sodium in the diet is not insignificant for the intestinal microbiota inhabiting the body. Recently, it was suggested that a *Bacteroides*/*Prevotella* ratio could be indicative of dietary excess of sodium since it decreases in response to a high-salt diet [148]. In rodents, such a diet decreases the diversity of the microbiota and promotes such taxons as *Christensenellaceae*, *Corynobacteriaceae*, *Lachnospiraceae*, *Ruminococcae,* and *Oscillospira* [34]. Nevertheless, from a health perspective, an undesirable consequence of a high salt intake is a depletion of *Lactobacillus*, which was observed in humans and animals [149,150]. The decrease in the abundance of this bacterial genus could lead to increased inflammatory status, and it has been suggested that the intestinal microbiota may significantly affect the sodium-induced inflammation resulting from the buildup of this electrolyte in the body [12]. Microbiota changes in response to increased salt intake seem to affect the production of interleukin-17, and through this, elevate the sodium reabsorption in the kidneys [151,152].

Recently, researchers gained interest in extracellular vesicles (EVs) produced by microbiota and their impact on the organism of host. Cells communicate using EVs, packing them with various loads, such as nucleic acids, proteins (including sodium transporters), lipids, and other molecules [153,154]. Bacterial EVs enter from the intestinal lumen to the bloodstream and are removed from the body together with urine. Some researchers have sequenced nucleic acids from EVs in human urine and concluded that information about the human microbiota sourced in this way could be more indicative of health and disease than the traditional faecal microbiota analysis [155,156].

In hypertension, the level of circulating and urinary EVs is elevated [157]. Microbiota-derived EVs could contribute to this. Some authors have suggested that microbiota may use EV to influence sodium absorption and utilisation in the body [158], but so far there is little research evidence to support this hypothesis. Nevertheless, some of the microbial EVs have already been researched as a potential antihypertensive medication. *Akkermansia muciniphila*-produced EVs, injected into spontaneously hypertensive rats, were found to prevent hypertension [159]. These EVs were also found to exhibit potent T-cell-mediated anti-inflammatory properties, protecting the liver and gut, which could explain their hypotensive effect [160]. Nevertheless, the potential impact of EVs produced by *A. muciniphila* on the bioavailability of dietary sodium requires further research.

Phenomena through which dietary fibre and microbiota may impact the absorption and retention of sodium in the human body are summarised in Table 2. Overall, the literature suggests that both the fibre and the microbiota profoundly impact sodium’s bioavailability. However, further research is required to translate this knowledge into practical applications for hypertension treatment.

## 7. Conclusions

Nowadays, lowering salt consumption is included in non-pharmacological interventions for the management of hypertension. Nevertheless, applying the existing guidelines to an effective practice poses a challenge. To help address the issue of excess sodium intake, there is a need for supporting interventions that would alter the fate of sodium upon ingestion and could be easily incorporated into daily life.

The findings of this review indicated that dietary fibre may reduce the bioavailability of sodium by retaining it in the faecal matter and decreasing its intestinal absorption through the binding of bile salts. In addition, dietary fibre affects the human intestinal microbiota, which may further impact the fate of sodium in the body. Certain microbial metabolites were shown to inhibit intestinal sodium transporters or ameliorate systemic inflammation, thereby reducing renal sodium reabsorption. In addition, the impact of bacterial EVs on sodium metabolism may be of interest for further research, especially as some of these entities demonstrated hypotensive potential in animal studies.

Consequently, interventions aimed at increasing dietary fibre intake and encouraging beneficial changes in microbiota could support the treatment of salt-sensitive hypertension. However, despite these promising discoveries, further studies enabling the establishment of effective, evidence-based interventions are required. Key priorities include identifying the optimal quality and quantity of dietary fibre, understanding the microbiota’s response to this food source, not only in terms of diversity but also in active metabolite production, and investigating how individual microbial communities respond to the antihypertensive medication. A precision nutrition approach using data on an individual’s microbiota, is a potential avenue for effective, personalised hypertension management that warrants further exploration.

## Figures and Tables

**Figure 1 microorganisms-13-01269-f001:**
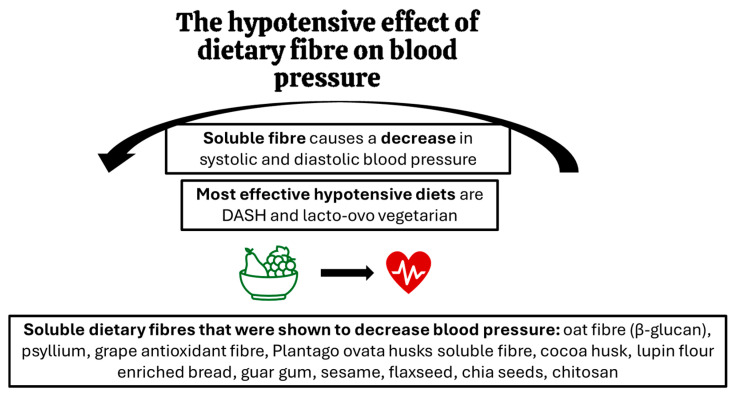
The effects of dietary fibre on blood pressure: key findings.

**Figure 2 microorganisms-13-01269-f002:**
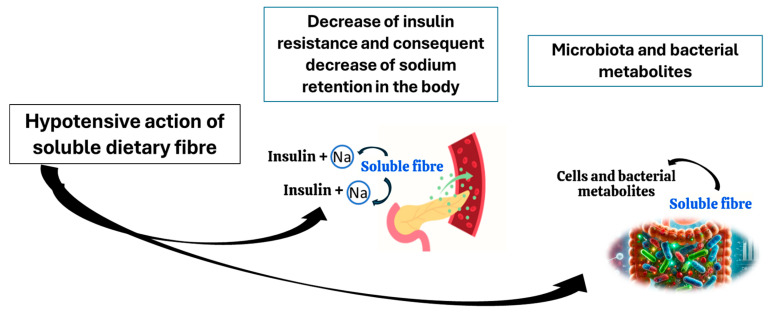
Potential mechanisms of hypotensive action of soluble fibre.

**Table 1 microorganisms-13-01269-t001:** Recommendations for sodium intake for adults.

Recommendation	Sodium Intake Value (g/Day)	Issuing Organisation	Reference
Safe and adequate intake	2	The European Food Safety Authority	[34]
Sufficient intake	1.5	French Agency for Food, Environmental and Occupational Health	[37]
Sufficient intake	1.5	Nordic Council of Ministers	[38]
Reference Nutrient Intake	1.6	Scientific Advisory Committee on Nutrition (UK)	[39]
Lower Reference Nutrient Intake	0.575		
Adequate intake	1.5	German Nutrition Society	[35]
Sodium standards for the population of the United States and Canada	1.5	National Academies of Sciences, Engineering, and Medicine	[36]
Adequate intake	1.5	Narodowy Instytut Zdrowia Publicznego–Państwowy Zakład Higieny (Poland)	[40]

**Table 2 microorganisms-13-01269-t002:** Interactions through which dietary fibre and microbiota (or its metabolites) may impact the fate of sodium in the human body.

Lowering Na^+^ Retention/Accession in the Body	Increasing Na^+^ Retention/Accession in the Body
Increased Na^+^ excretion with the increase in dietary fibre intake	Increasing dietary fibre intake increases the number of Na^+^ transporters in the intestines
Bondage of bile salts by dietary fibre	Supplementation with short-chain fatty acids may increase the number of Na^+^ transporters in the intestines
Inhibition of Na^+^ transporters by some of the bacterial metabolites (e.g., several known bacterial toxins)	Inflammation associated with the dysbiosis of microbiota may increase Na^+^ reabsorption in the kidneys
Anti-inflammatory action of some of the microbiota-derived extracellular vesicles (e.g., from *Akkermansia muciniphila*)

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
