# Peer review of "The Role of Intestinal Microbiota and Dietary Fibre in the Regulation of Blood Pressure Through the Interaction with Sodium: A Narrative Review"

_microorganisms, 2025, doi:10.3390/microorganisms13061269_

Round 1
Reviewer 1 Report
Comments and Suggestions for Authors
Thank you very much for allowing me to review the review article entitled “microorganisms-3581791_The role of intestinal microbiota and dietary fibre in the regulation of blood pressure through the interaction with sodium, a narrative review. ”, which is included in the section “Gut Microbiota” of the Special Issue “Gut Microbiota and Nutrients, 2nd Edition ”.
The aim of this review is to concisely summarise the mechanisms by which dietary fibre and the gut microbiota may influence blood pressure, with a particular focus on their interaction with sodium—an essential dietary component implicated in the pathogenesis of hypertension.
Comments:
The abstract serves as the main presentation of the article and remains freely accessible across all platforms. Therefore, it should provide a comprehensive overview of the content. In this context, the abstract should include not only the hypothesis and objective of the review, but also key methodological elements such as the number of articles analysed, the time frame covered, the databases consulted, the principal findings, and the conclusions. For these reasons, I strongly recommend that the abstract be rewritten to reflect this.
The introduction appropriately outlines the topic of interest, supported by relevant literature, and concludes by clearly stating the objective of the review.
The results are organised thematically, a strategy that enhances the clarity and structure of the review.
However, while the results section includes a discussion of various studies that examine the relationship between sodium intake and hypertension, as well as the cardiovascular benefits of reducing salt consumption, it does not sufficiently address the underlying mechanisms—despite this being the stated aim of the review. Similarly, the section presenting institutional recommendations on salt intake for different age groups and populations, including national legislation, also lacks mechanistic insight.
Another section discusses the pathophysiology of hypertension at the renal level, its effects on various organs, and the role of high-sodium diets in increasing risk—thereby supporting the rationale for reduced sodium intake.
Subsequent sections examine the impact of dietary fibre on hypertension via modulation of the gut microbiota, and finally, the potential influence of dietary fibre and microbiota on systemic homeostasis. This final section most directly addresses the stated objective of the review, whereas the other sections are somewhat tangential. I therefore suggest that the authors more clearly focus the manuscript on their central research aim.
The conclusion currently reads more as a summary than as a direct response to the review’s objective. It should be rewritten to explicitly answer the central research question.
In the current scientific landscape, review articles play a crucial role in synthesising the rapidly expanding body of literature, thereby aiding researchers and clinicians in keeping abreast of developments in their fields. Even in the case of a narrative review, it is essential to include a minimum methodology section detailing the time period covered, databases consulted, and inclusion/exclusion criteria used. These elements are fundamental for establishing the review's rigour and for facilitating its comparison with future work in the field.
In general, I consider this to be an interesting and relevant study. However, the manuscript lacks a clear focus on the stated objective and is currently missing essential methodological detail.
Author Response
Comments 1:
Thank you very much for allowing me to review the review article entitled “microorganisms-3581791_The role of intestinal microbiota and dietary fibre in the regulation of blood pressure through the interaction with sodium, a narrative review. ”, which is included in the section “Gut Microbiota” of the Special Issue “Gut Microbiota and Nutrients, 2nd Edition ”.
The aim of this review is to concisely summarise the mechanisms by which dietary fibre and the gut microbiota may influence blood pressure, with a particular focus on their interaction with sodium—an essential dietary component implicated in the pathogenesis of hypertension.
Response 1:
Thank you for reading our manuscript and contributing to improving its quality. We appreciate all the comments and hope that the changes that were made address them adequately.
Comments 2:
The abstract serves as the main presentation of the article and remains freely accessible across all platforms. Therefore, it should provide a comprehensive overview of the content. In this context, the abstract should include not only the hypothesis and objective of the review, but also key methodological elements such as the number of articles analysed, the time frame covered, the databases consulted, the principal findings, and the conclusions. For these reasons, I strongly recommend that the abstract be rewritten to reflect this.
Response 2:
Thank you very much for your comment. We have now rewritten the abstract, leaving only the first paragraph as in the previous version.
Comments 3:
The introduction appropriately outlines the topic of interest, supported by relevant literature, and concludes by clearly stating the objective of the review.
Response 3:
Thank you very much for your kind comment.
Commends 4:
The results are organised thematically, a strategy that enhances the clarity and structure of the review.
However, while the results section includes a discussion of various studies that examine the relationship between sodium intake and hypertension, as well as the cardiovascular benefits of reducing salt consumption, it does not sufficiently address the underlying mechanisms—despite this being the stated aim of the review. Similarly, the section presenting institutional recommendations on salt intake for different age groups and populations, including national legislation, also lacks mechanistic insight.
Response 4:
We appreciate the comment. We have now added a paragraph in section 3 (formerly section 2), lines 108-115, on mechanisms through which sodium intake affects blood pressure regulation and provided references that describe these mechanisms in detail. This paragraph reads:
“Excessive sodium consumption, along with impaired renal sodium excretion, leads to an increased blood pressure level. As a result, the peripheral vascular resistance is elevated. Anatomic remodelling of small resistance arteries is accompanied by vascular endothelial inflammation and structural changes of large elastic arteries (increase of arterial stiffness) [14,15]. In addition to vascular dysfunction, the pathophysiology of sodium-sensitive blood pressure includes many factors, such as genetic, neuro-endocrine (renin-angiotensin-aldosterone system, natriuretic peptides, etc.), and autonomic nervous system imbalance (increased sympathetic activity) [16].”
Comments 5:
Another section discusses the pathophysiology of hypertension at the renal level, its effects on various organs, and the role of high-sodium diets in increasing risk—thereby supporting the rationale for reduced sodium intake.
Response 5:
Thank you. We agree with the comment.
Comments 6:
Subsequent sections examine the impact of dietary fibre on hypertension via modulation of the gut microbiota, and finally, the potential influence of dietary fibre and microbiota on systemic homeostasis. This final section most directly addresses the stated objective of the review, whereas the other sections are somewhat tangential. I therefore suggest that the authors more clearly focus the manuscript on their central research aim.
Response 6:
Thank you very much for drawing attention to this issue. We have now reduced the amount of text in former section 2 (now 3), changing a major part of the text, which referred to recommendations for sodium intake, into a table and merged it with former section 2, while removing repetitions (first paragraph of former section 3).
Comments 7:
The conclusion currently reads more as a summary than as a direct response to the review’s objective. It should be rewritten to explicitly answer the central research question.
Response 7:
Thank you for your comment. We have now amended the conclusions, leaving only two first sentences of original version.
Comments 8:
In the current scientific landscape, review articles play a crucial role in synthesising the rapidly expanding body of literature, thereby aiding researchers and clinicians in keeping abreast of developments in their fields. Even in the case of a narrative review, it is essential to include a minimum methodology section detailing the time period covered, databases consulted, and inclusion/exclusion criteria used. These elements are fundamental for establishing the review's rigour and for facilitating its comparison with future work in the field.
Response 8:
Thank you very much for pointing this out. We agree, certainly most of the reviews, including narrative reviews, published nowadays, include a methodology section. We have now added such a section to our manuscript (section 2, lines 79-90). It reads: “The literature search included peer-reviewed articles and governmental or organisational websites presenting current dietary recommendations for sodium intake, published online in English up to March 2025. While we have focused on recent references (< 5 years), some older sources of fundamental meaning to the field or unique research value were also included.
The search was conducted using the following databases: Google, GoogleScholar, PubMed, Web of Science, with a combination of following keywords: “sodium”, “microbiota”, “microbiome”, “dietary fibre”, “diet”, “hypertension”, “sodium sensitivity”, “recommendations for sodium intake”, “bioavailability of sodium”. In total, over a thousand titles were screened for suitability, and finally, 160 references were used as a basis for this article.”
Comments 9:
In general, I consider this to be an interesting and relevant study. However, the manuscript lacks a clear focus on the stated objective and is currently missing essential methodological detail.
Response 9:
We appreciate your insights. We hope that the changes that were made in response to previous comments addressed these issues.
Reviewer 2 Report
Comments and Suggestions for Authors
Summary:
The study titled “The role of intestinal microbiota and dietary fibre in the regulation of blood pressure through the interaction with sodium, a narrative review” discusses the role of gut microbiota and dietary fiber in regulating blood pressure, focusing on their interaction with sodium. The authors highlight that dietary fibers may reduce sodium bioavailability and support beneficial microbial populations, which could impact hypertension. They call for further research to explore fiber and microbiota-based precision nutrition approaches in hypertension management.
The manuscript is logically structured and appropriately segmented. Each section flows coherently into the next. The literature used is current and comprehensive, with numerous references from 2023–2025, reflecting the latest research in the field. This greatly enhances the review’s credibility and scientific value.
Observations:
Overall, the English language is of high quality and meets the standards of scientific writing. The manuscript is written in a clear and scholarly tone. Minor grammatical and stylistic issues were identified (e.g., subject-verb agreement, unnecessary commas, and occasional word choices), but these do not impede understanding and can be easily corrected during proofreading.
Line |
Original text |
Correction |
23 |
that fibre exhibits |
that fibre has been shown to exhibit |
35 |
Hypertension is an illness, as well as a serious risk factor... |
Hypertension is a condition and a serious risk factor... |
60 |
sellection |
selection |
90 |
so little sodium |
very little sodium |
136 |
Nutrtional |
Nutritional |
149 |
is 1500 mg/day |
are 1500 mg/day |
606 |
need to be assessed |
needs to be assessed |
607 |
hypertension medication |
antihypertensive medication |
608 |
approach, with the use of data on an individual’s microbiota is... |
approach using data on an individual’s microbiota is... |
I recommend this manuscript for publication after minor revisions, particularly to correct the identified grammatical errors and to slightly refine the discussion around mechanisms with inconsistent evidence.
Author Response
Comments 1:
The study titled “The role of intestinal microbiota and dietary fibre in the regulation of blood pressure through the interaction with sodium, a narrative review” discusses the role of gut microbiota and dietary fiber in regulating blood pressure, focusing on their interaction with sodium. The authors highlight that dietary fibers may reduce sodium bioavailability and support beneficial microbial populations, which could impact hypertension. They call for further research to explore fiber and microbiota-based precision nutrition approaches in hypertension management.
The manuscript is logically structured and appropriately segmented. Each section flows coherently into the next. The literature used is current and comprehensive, with numerous references from 2023–2025, reflecting the latest research in the field. This greatly enhances the review’s credibility and scientific value.
Response 1:
Thank you for reading our manuscript and contributing to improving its quality. We appreciate all the comments and hope that the changes that were made, address them adequately.
Comments 2:
Observations:
Overall, the English language is of high quality and meets the standards of scientific writing. The manuscript is written in a clear and scholarly tone. Minor grammatical and stylistic issues were identified (e.g., subject-verb agreement, unnecessary commas, and occasional word choices), but these do not impede understanding and can be easily corrected during proofreading.
Line |
Original text |
Correction |
23 |
that fibre exhibits |
that fibre has been shown to exhibit |
35 |
Hypertension is an illness, as well as a serious risk factor... |
Hypertension is a condition and a serious risk factor... |
60 |
sellection |
selection |
90 |
so little sodium |
very little sodium |
136 |
Nutrtional |
Nutritional |
149 |
is 1500 mg/day |
are 1500 mg/day |
606 |
need to be assessed |
needs to be assessed |
607 |
hypertension medication |
antihypertensive medication |
608 |
approach, with the use of data on an individual’s microbiota is... |
approach using data on an individual’s microbiota is... |
I recommend this manuscript for publication after minor revisions, particularly to correct the identified grammatical errors and to slightly refine the discussion around mechanisms with inconsistent evidence
Response 2:
Thank you very much for this comment. We have now applied all suggested language corrections and rephrased a sentence formerly in lines 497-498 (“These results underscore the importance of fibre rather than purely SCFA in blood pressure regulation.”), now 464-465 (“These results highlight that fiber intake, rather than SCFA supplementation alone, may be more important for blood pressure regulation.”).
Major changes to the text, particularly in former section 2 and the conclusions, were also applied. Thus, we hope that this comment was addressed adequately.
Round 2
Reviewer 1 Report
Comments and Suggestions for Authors
I have thoroughly reviewed the manuscript “microorganisms-3581791_The role of intestinal microbiota and dietary fibre in the regulation of blood pressure through the interaction with sodium, a narrative review.” once again, as well as the authors’ responses to the reviewers’ comments. I recognize the considerable effort they have made to address the concerns raised, reorganize certain sections, and include a detailed methodology to enhance the clarity and comprehensibility of their work.
For all these reasons, I would like to commend the authors on the significant improvements and the high quality of the revised manuscript.
Reducing salt intake is essential in managing hypertension, although its practical implementation remains challenging. This review highlights the potential role of dietary fibre in lowering sodium bioavailability by reducing absorption and modulating gut microbiota. Microbial metabolites may inhibit sodium transporters and attenuate inflammation, contributing to reduced renal sodium reabsorption. Increasing fibre intake and promoting beneficial microbiota shifts may support hypertension management. Nonetheless, further research is needed to establish effective, evidence-based, and personalized interventions.